# Evaluating the Inclusion of Cold-Pressed Rapeseed Cake in the Concentrate for Dairy Cows upon Ruminal Biohydrogenation Process, Ruminal Microbial Community and Milk Production and Acceptability

**DOI:** 10.3390/ani11092553

**Published:** 2021-08-30

**Authors:** Idoia Goiri, Izaro Zubiria, Jose Luís Lavín, Hanen Benhissi, Raquel Atxaerandio, Roberto Ruiz, Nerea Mandaluniz, Aser García-Rodríguez

**Affiliations:** Campus Agroalimentario de Arkaute s/n, NEIKER-Basque Institute for Agricultural Research and Development, Basque Research and Technology Alliance (BRTA), 01192 Vitoria-Gasteiz, Spain; igoiri@neiker.eus (I.G.); izaro23@hotmail.com (I.Z.); jllavin@neiker.eus (J.L.L.); hanening@gmail.com (H.B.); ratxaerandio@neiker.eus (R.A.); rruiz@neiker.eus (R.R.); nmandaluniz@neiker.eus (N.M.)

**Keywords:** oilseed, alternative feedstuffs, palm oil, soybean meal, dairy cattle

## Abstract

**Simple Summary:**

Soybean meal and palm oil are important protein and energy sources in European livestock production, respectively. In the next decades, the demand for these feedstuffs is supposed to increase as the world population and its demand for meat and dairy products increases. Alternatives to replace those feedstuffs are necessary. It is necessary to promote low-input, local, and circular production systems: in this sense, adopting feeding systems that use cheaper and local alternative feedstuffs represents a good strategy. One of them is the product obtained after a simple pressing process for the production of biofuels. Cold pressing usually produces oil that can be used as biofuel and a cake rich in fat and with high oil quality but with lower protein content than the conventionally solvent-extracted cakes. Therefore, the availability of cold-pressed cakes can represent an example of integration between the industry and livestock production. The objective of this study was to assess the suitability of formulating cold-pressed rapeseed cake (CPRC) in a dairy cows’ concentrate as a substitute for conventional feedstuffs. Feeding CPRC has the advantage of slightly improving the milk fatty acid profile and consumer acceptance. In conclusion, CPRC can replace conventional feedstuffs without detrimental effects on milk production and composition.

**Abstract:**

The aim of this trial was to assess the effect of feeding a concentrate including cold-pressed rapeseed cake (CPRC) on productive performance, milk quality and its sensory properties, ruminal biohydrogenation, and bacterial communities. Eighteen cows were paired, and two experimental diets (control vs. CPRC) were distributed within the pair. Concentrates were iso-energetic and iso-proteic and contained similar amounts of fat. The average days in milk, milk yield, and body weight of the animals were (mean ± SD) 172 ± 112 d, 585 ± 26 kg, and 25.4 ± 6.2 kg/d, respectively. The experiment lasted for 10 wk. Feeding CPRC resulted in lower ruminal saturated (*p* < 0.001) and higher monounsaturated (*p* = 0.002) fatty acids. Feeding CPRC increased *Ruminococcus*, *Prevotella*, and *Entodinium* but decreased *Blautia*; *p-75-a5;* undefined genera within orders Clostridiaceae and RF39 and within families Christensenellaceae, Lachnospiracease, and Ruminococcaceae; and fungi from the phylum neocallimastigomycota. The milk fatty acid profile was characterized by a lower n6:n3 ratio (*p* = 0.028). Feeding CPRC did not affect the milk yield, milk quality, or fat corrected milk (*p* > 0.05). Feeding CPRC improved the overall milk acceptability (*p* = 0.047). In conclusion, CPRC affected some microbial taxa, modified the biohydrogenation process, and improved the milk fatty acid profile and consumer acceptance without detrimental effects on milk production and composition.

## 1. Introduction

Soybean meal and palm oil are important protein and energy input sources in European livestock production. In the next decades, the demand for these feedstuffs is supposed to increase as the world population and its demand for meat and dairy products increases [1]. The creation of massive plantations to produce either soybean meal [2] or palm oil [3] means that wild forests have been replaced with monocultures. Burning of forests to make space for soybean or palm also has social and environmental impacts [2]. Deforestation removes the local economy of these regions, reduces biodiversity, and pollutes the environment, being recognized as a major contributor to the emission of greenhouse gases [2]. In addition, the impact of the transportation of these feedstuffs from their origin has to be kept in mind in the context of the actual need for more sustainable farming systems.

The EU has been supporting farmers to adopt or maintain practices that contribute to fulfill environmental and climate goals through the “green direct payment” (or “greening”), which rewarded, among others, crop diversification. This policy is expected to continue through the agricultural practices included in the eco-schemes of the new CAP reform. In recent decades in Europe, this fact has led to an increase in landing of different oilseeds, such as sunflower or rapeseed. In this context of increased oilseed landing, alternative uses have been proposed. One example has been a simple pressing process for the production of biofuels in local areas using cold-pressing. Cold presses are normally mechanically operated and often involve a screw device that is tightened against the paste to extract the oils. In this process, therefore, neither solvents nor heat are applied to help in the extraction. This process usually produces oil that can be used as biofuel and a cake rich in fat and with high oil quality but with lower protein content than the conventionally solvent-extracted cakes. As a consequence, an innovative local agricultural production chain based on the cultivation of oilseeds has been developed. Livestock production systems in the same area require protein and energy supplements from the market. Therefore, the availability of cold-pressed cakes can represent an example of integration between the industry and local livestock production.

Feeding studies with cold-pressed rapeseed cake (CPRC) have shown that its use as a lipid supplement in ruminant diet is susceptible to reducing the extent of ruminal biohydrogenation (BH) and to modifying the ruminal fatty acid (FA) profile towards reduced saturated FA content and increased mono- and polyunsaturated FA levels, as was documented in meat [4], dairy sheep milk [5,6], and cheese [7]. These changes occur with no detrimental effect on the rumen in vitro fermentation process [8,9], apparent digestibility [10], or productive performance [5,6], presenting this cake as a very promising alternative to soybean cake and palm oil. However, there is a lack of studies covering the potential use of CPRC on dairy cow diets’ and its effect on milk production and quality.

The effect of dietary lipids on nutrients degradability and digestibility as well as on animal products FA profile is well known. These relationships are mainly mediated by the toxicity of unsaturated fatty acid (UFA) on many microorganisms, especially fibrolytic bacteria [11]. Nevertheless, no recent studies have elucidated the specific effect of CPRC, rich in UFA, on ruminal populations so far. To better understand how CPRC affects the ruminal ecosystem, sequencing methods are promising to be implemented, offering detailed information about the microbial complexity and functionality.

Therefore, in the present study, we used locally produced oil-rich CPRC in the formulation of a concentrate for dairy cows and we hypothesize that the oil present in this cake, rich in UFA, could modify the ruminal microbial communities and therefore ruminal biohydrogenation process, leading to a shift in milk FA profile towards an improved n3:n6 ratio. Moreover, we also expect a change in milk sensory characteristics due to the expected more unsaturated milk FA profile [12]. For this reason, the objective of the current trial was to assess the effect of feeding a concentrate including CPRC on productive performance, milk composition and its sensory properties, ruminal biohydrogentation process, and bacterial communities.

## 2. Materials and Methods

### 2.1. Animals and Treatments

The trial was conducted at the experimental research farm of Fraisoro Farm School (Zizurkil, Spain). Cows were in loose housing conditions. A total of 18 cows were used; 10 Holstein (H) and 8 Brown Swiss (BS). The average days in milk (DIM), body weight (BW), number of parity, and milk yield of the cows before starting the experiment were (mean ± SD) 172 ± 112 d, 585 ± 26 kg, 1.9 ± 1.1, and 25.4 ± 6.2 kg/d, respectively. Animals were paired by taking into account the breed, number of parity, DIM, and mean milk yield during a 2-week covariate period. At the end of the covariate period, cows were randomly assigned (within pair) to the CTR or CPRC concentrate for the experimental trial. Concentrates were fed for 10 wk; the first 2 wk were for adaptation to the diets, and during the last 8 wk, measurements were taken. Concentrates were formulated to be iso-proteic and iso-energetic and to provide similar amounts of fat, following the NRC [13] recommendations (Table 1). Table 2 shows the FA profile of the two experimental concentrates. Cows within a pair received the same quantity of concentrate (7 ± 1.3 kg/d), but they had ad libitum access to a basal forage ration. Concentrate was offered in individual buckets three times per day.

Cows were milked with an automatic milking system (AMS, DeLaval, 2004) machine with free access to the AMS for 22.5 h/d (1.5 h for cleaning of the system). Milking intervals were set to a minimum of 6 h from the previous milking. Nevertheless, if a milking failure occurred, cows would be granted permission to be milked again immediately. During the day, for any particular cow, when the time passed since last milking was more than 12 h, the cow would be forced to visit the AMS.

### 2.2. Sampling and Measurements

Daily milk production was recorded individually at each milking by the AMS. On the last day of the covariate period and during weeks 2, 4, 6, 8, and 10 of the experimental period, milk samples were taken from the AMS at each milking and stored with azidiol (3.3 mL/L) at 4 °C for fat, protein, and lactose determination (ILL, Lekunberri, Spain). Additional milk samples were taken at each milking on weeks 4 and 9. Then, these milk samples were bulked by animal and day on a milk production basis and were stored at −20 ± 5 °C with azidiol (3.3 mL/L) for FA composition analysis (LIGAL, Mabegondo, Spain). Offered basal forage and orts were sampled on a daily basis, and concentrate feeds were sampled weekly to characterize their chemical composition.

In week 9 of the experimental period, rumen samples were collected over two consecutive days for analysis of the FA profile and for DNA extraction for the study of the ruminal microbial community. Sampling was performed at 00:00 and 12:00 h on the first day and at 06:00 and 18:00 h on the second day. Ruminal samples were collected from each dairy cow using a stomach tube (18 mm diameter and 160 cm long) connected to a mechanical pumping unit (Vacuubrand ME 2SI, Wertheim, Germany). The ruminal content was filtered using four layers of sterile gases. For FA profile analyses, a 100 mL pool was made for each cow with 25 mL of the liquid fraction of each ruminal extraction. For DNA extraction in the study of the ruminal microbial community, another 100 mL of each ruminal extraction was saved into a container. All samples were immediately stored frozen at −20 ± 5 °C until analysis.

In the last week of the trial, a composite milk sample (36 kg) from each treatment was taken in stainless steel milk cans for subsequent sensory analysis.

### 2.3. Sample Handling and Laboratory Procedures

#### 2.3.1. Feed

Basal forage and concentrates were dried in a forced-air oven (60 °C/48 h) and ground through a 1 mm sieve. The samples were analyzed for dry matter (method 934.01) and N (method 984.13) content following AOAC [14]. Neutral detergent fiber was determined by the method of [15] with the use of an alpha amylase but without sodium sulfite and was expressed free of ash. The acid detergent fiber, expressed exclusive of residual ash, was determined by the method of [16]. The ether extract content was determined without hydrolysis by the automated soxhlet method (Soxtec System HT 1043 Extraction Unit, Madrid, Spain) using hexane for 6 h as solvent. The starch content was measured by polarimetry [17].

Fatty acid methyl esters (FAME) of lipid in both concentrates were prepared in a 1-step extraction-trans-esterification procedure using chloroform and 20 mL/L sulfuric acid in methanol [18]. Methyl esters were separated and quantified with a gas chromatograph (Agilent 7890A GC System, Santa Clara, CA, USA) equipped with a flame-ionization detector, a 100 m fused silica capillary column (0.25 mm i.d., 0.2-μm film thickness; CP-SIL 88, CP7489, Varian Ibérica S.A., Madrid, Spain), and hydrogen as the carrier gas (207 kPa, 2.1 mL/min). The total FAME profile in a 2 μL sample volume at a split ratio of 1:50 was determined using the temperature gradient program described in [18]. Peaks were identified based on retention time comparisons with commercially available standard FAME mixtures (Nu-Chek Prep., Elysian, MN, USA; and Sigma-Aldrich, Madrid, Spain).

#### 2.3.2. Rumen Fatty Acid Profile Analysis

Rumen FA profile determinations were performed as described in [19]. Briefly, lipid in 200 mg freeze-dried rumen samples was extracted with a mixture of hexane and isopropanol (3:2, vol/vol; [18]) and converted to FAME by sequential base-acid catalyzed transesterification [20]. The total FAME profile was analyzed by gas chromatography using the same chromatograph and temperature gradient program utilized for the analysis of FA in feeds, but isomers of 18:1 were further resolved in a separate analysis under isothermal conditions at 170 °C [18]. Peaks were identified based on retention time comparisons with the same FAME mixtures used for the analysis of feeds and other commercially available standards (from Nu-Chek Prep.; Sigma-Aldrich; and Larodan, Solna, Sweden), cross referencing with chromatograms reported in the literature (e.g., [18,20]), and based on a comparison with reference samples for which the FA composition was determined based on gas chromatography analysis of FAME and gas chromatography-mass spectrometry analysis of corresponding 4,4-dimethyloxazoline derivatives [20].

#### 2.3.3. Rumen DNA Extraction

Rumen samples were thawed for 10 h at refrigeration temperature (5 ± 3 °C) and squeezed using four layers of sterile gases to separate between solid (particle size smaller than the diameter of the stomach tube) from liquid digesta phases. Liquid digesta phase was separated into planktonic organisms and bacteria associated with the liquid fraction. The solid phase was separated between associated and adherent fractions following the methodology described in [21]. The four fractions obtained were lyophilized and composited to obtain a unique sample with the four fractions represented proportionally (on dry matter basis). DNA extraction was performed using the commercial Power-Soil DNA Isolation kit (Mo Bio Laboratories Inc, Carlsbad, CA, USA) following manufacturer’s instructions. The extracted DNA was subjected to paired-end Illumina sequencing of the V4 hypervariable region of the 16S rRNA [22] and of the V7 region of the 18S rRNA genes. The libraries were generated by means of Nextera kit. The 250 bp paired-end sequencing reactions were performed on a MiSeq platform (Illumina, San Diego, CA, USA). The bacterial and archaeal communities were grouped as OTUs (Operational Taxonomic Units) based on 16S rRNA similarities and protozoal and fungi on 18S rRNA similarities. Data processing was performed using QIIME (v.1.9.0): Quantitative Insights Into Microbial Ecology software package [23]. Sequences were clustered as operational taxonomic units (OTUs) of 97% similarity using UCLUST [24]. OTUs were checked for chimeras using the RDP gold database and assigned bacterial and archaeal 16S RNA taxonomy using the Greengenes database [25], whereas protozoal and fungi 18S rRNA genes were aligned against the 18S SILVA database [26]. Alpha and beta diversity metrics were calculated using the QIIME pipeline.

#### 2.3.4. Milk

Milk fat, protein, and lactose contents were analyzed by near-infrared spectroscopy (Foss System 4000, Foss Electric, Hillerød, Denmark; ILL, Lekunberri, Spain). To analyze the milk FA profile, milk fat extraction was carried out according to ISO 14156 [27], methylated according to ISO 15884 [28], and analyzed using gas chromatography. The upper phase was injected into a gas chromatograph (Varian 3800) equipped with a capillary column (Cp-sil 88 to over 50 m) and the FID detector. Working conditions were set according to the standard [29]. The carrier gas, nitrogen with a pressure of 14 psi, was used, and the injector temperature was 250 °C. Temperature program proposed by Kramer et al. [30] was used: 4 min at 45 °C, then an increase in temperature of 13 °C per minute up to a temperature of 175 °C (27 min), and an increase in temperature of 4 °C per minute up to 215 °C (35 min).

#### 2.3.5. Pasteurized Milk Perceptibility and Sensory Properties

Raw milk was pasteurized at 72 °C for 30 s using a continuous plate heat exchanger (ATA Tecnología Alimentaria, Irun, Spain). A triangle test was performed to analyze the consumers’ ability to distinguish differences between samples for the attributes of appearance, flavor, odor, texture, and overall acceptability. Forty untrained panelists evaluated four milk samples per treatment in private booths. The panelists were served 2 sets of samples in which the reference was either milk from the CTR or CPRC diet. In every set, one sample was identical to the reference and one was different. For each sample set, the panelists had to identify the sample that tasted the same as the reference. The acceptance test was carried out using a non-trained sensory panel of women and men, regular consumers of cow milk. A 9-point line scale was used, with 1 being the lowest and 9 being the highest score, for each of the measured attributes.

### 2.4. Calculations and Statistical Analysis

Milk fat, protein, and lactose concentrations were calculated as weighted average of milking data: 3.5% fat corrected milk (FCM) was calculated as 0.4318M + 16.23F, with M being milk production (kg) and F being milk fat (kg).

For statistical purposes, each cow was considered the experimental unit (*n* = 18). Milk yield, FCM, milk fat and protein contents, and milk fat and protein yield were analyzed by a MIXED model for repeated measures using the MIXED procedure of SAS software [31], assuming a covariance structure fitted on the basis of Schwarz’s Bayesian information model fit criterion.
$$Y_{jklmn}=\;µ\;+Cov_{j}+T_{k}+P_{l}+W_{m}+C{\left(P\right)}_{n}+\varepsilon_{jklmn}$$
where *Y* is the dependent variable, µ is the mean values for each treatment, *Cov* is the initial record (covariate), *T* is the fixed effect of the concentrate used, *P* is the fixed effect of the pair, *W* is the fixed effect of the week (week 3–week 10), *C*(*P*) is the random effect of cow within pair, and ε is the residuals. Least squares means for treatments are reported.

Rumen FA concentrations were averaged by cow. The rumen and milk FA concentrations were analyzed using the previous statistical model but without considering covariate or repeated measures. The sensorial data (*n* = 40) were analyzed using the previous statistical model but without considering the effect of the week. Treatment means were separated using a Tukey test except for rumen and milk FA profile, where Bonferroni adjustment was used. Significant effects were declared at *p* ˂ 0.05.

Relative abundances (RA) of bacterial and eukaryote taxa at the phylum, family, and genus level were analyzed using the MIXED procedure [31], according to the following model:Y_jkl_ = µ + T*_j_* + P*_k_* + C(P)*_l_* +ε*_jkl_*
where Y is the dependent variable, µ is the mean values for each treatment, T is the fixed effect of the concentrate used, P is the fixed effect of the pair, C(P) is the random effect of cow within pair, and ε is the residuals. Residuals were checked for normality with either the Shapiro–Wilk or Kolmogorov–Smirnov tests using [31], and the data were transformed (log, square-root, and reciprocal transformation) when necessary until the residuals followed a normal distribution.

Significant differences between experimental groups’ bacterial and eukaryote community composition were analyzed by analysis of dissimilarity (ADONIS) with 999 permutations. The significant fold changes of the OTUs were tested using DESeq2 [32] and filtered by the false discovery rate value.

To investigate the associations between ruminal FAs and bacterial taxa, a regularized canonical correlation analysis (rCCA) was carried out with the package mixOmics (v6.6.2) [33] in R (v3.5.1) [34]. To perform the rCCA analysis, the correlation values between the RA of bacterial genera and each ruminal FA proportions were computed to calculate a similarity matrix. A clustered image map was inferred using a similarity matrix obtained from the rCCA. A threshold of R = 0.45 was used to obtain the relevant components.

## 3. Results

### 3.1. Rumen Fatty Acid Composition

Diet containing CPRC changed the rumen saturated fatty acid (SFA) profile (Table 3). The main changes were significant decreases in the proportions of C12:0 (*p* < 0.001), C14:0 (*p* = 0.019), and C16:0 (*p* < 0.001) and increases in the proportions of C17:0 (*p* = 0.005), C18:0 (*p* = 0.007), C19:0 (*p* = 0.020), C20:0 (*p* < 0.001), and C22:0 (*p* < 0.001). The total SFA was lower in the CPRC experimental group (*p* < 0.001).
Escriba aqui la ecuación.

The diet containing CPRC resulted in an increase in total rumen MUFA (*p* = 0.002), cis MUFA (*p* = 0.028), and trans MUFA (*p* = 0.002; Table 4). Shifts were mainly characterized by an increase in the proportions of C18:1 cis-9 (*p* = 0.028), C18:1 trans-11 (*p* = 0.005), and C18:1 trans-13-14 (*p* = 0.020) but without changing the C18:1 trans-10/trans-11 ratio (*p* = 0.377).

The total ruminal PUFA contents were not affected (*p* = 0.829) by the diet containing CPRC (Table 4). The experimental concentrate with CPRC did not affect C18:2 cis-9 trans-11 CLA, C18:2 trans-9 cis-11 CLA, C18:2 trans-11 trans-13 CLA, and C18:2 trans-10 cis-12 CLA contents (*p* > 0.05). As a consequence, the diet with CPRC did not result in increased CLA proportions (*p* = 0.430). The proportions of long-chain *n*-3 PUFA (*p* = 0.157) and *n*-6 PUFA (*p* = 0.803) were not altered in the CPRC experimental group. As a consequence, the diet with CPRC did not alter the *n*-6:*n*-3 ratio (*p* = 0.263).

### 3.2. Ruminal Microbial Community

The main phyla were Bacteroidetes (50.7%) and Firmicutes (33.2%). Within the Bacteroidetes, the most abundant families were Prevotellaceae (42.4%), undefined families within the order of the Bacteroidales (4.2%), and (Paraprevotellaceae) (1.2%). The dominant families of Firmicutes were Lachnospiraceae (11.4%), Ruminococcaceae (7.3%), undefined families within order of the Clostridiales (7.3%), and Veillonellaceae (5.5%) (Figure 1).

The experimental concentrate with CPRC did not influence the bacterial or Eukaryote species richness as expressed by different diversity indices, such as chao1 or Shannon (Table 5). The beta diversity and the statistical test performed with ADONIS revealed no differences in bacterial (*p* = 0.186) community and a tendency in eukaryote (*p* = 0.063) community between experimental concentrates.

Among the different bacterial phyla (Appendix A), the diet containing CPRC only significantly decreased Tenericutes (*p* = 0.038) and tended to decrease Plantomycetes (*p* = 0.056).

At the family level (Appendix A), the diet with CPRC decreased RA of the undefined families within the order Clostridiales (*p* = 0.016), Christensenellaceae (*p* = 0.033), and the undefined families within the order RF39 (*p* = 0.024) and tended to decrease RA of Lachnospiraceae (*p* = 0.08), Ruminococcaceae (*p* = 0.08), and Pirellulaceae (*p* = 0.055).

At the genus level (Appendix A), the diet with CPRC increased RA of *Ruminococcus* (*p* = 0.047) and decreased RA of *Blautia* (*p* = 0.045); *p*-75-a5 (*p* = 0.013); undefined genus within the orders Clostridiales (*p* = 0.016) and RF39 (*p* = 0.024); and those within families Christensenellaceae (*p* = 0.033), Lachnospiraceae (*p* = 0.011), and Ruminococcaceae (*p* = 0.034) and tended to decrease Clostridium (*p* = 0.098), Shuttleworthia (*p* = 0.0506), Pyramidobacter (*p* = 0.072), and undefined genus within the family Pirellulaceae (*p* = 0.055).

At the OTU level, the OTU belonging to the genera *Prevotella* and *Ruminococcus* were enriched when the animals were fed the experimental concentrate with CPRC, whereas in the ruminal content of the animals fed the CTR concentrate, an enrichment in OTU of undefined genera within the order Clostridiales was observed (Figure 2).

The most abundant Eukaryote phyla (Figure 3) in both experimental groups (CPRC and CTR) were Ciliophora (42 and 34%), Ascomycota (33% and 32%), and neocallimastigomycota (4 and 11%).

Among the different Eukaryote phyla (Appendix A), diets with CPRC only significantly decreased neocallimastigomycota (*p* = 0.016) compared with the control (Figure 4).

Regarding the genera belonging to Ciliophora phylum, diets containing CPRC only significantly increased the RA of genus *Entodinium* (*p* = 0.039; Table 6). At the OTU level (Appendix A), some OTUs belonging to the genera *Entodinium*, undefined genus within subclass Trichostomatia, and *Ophryoscolex* were enriched in the rumen of animals fed the experimental concentrate with CPRC, whereas in the ruminal content of the animals fed the CTR concentrate, enrichments in OTUs of the undefined genus within subclass Haptoria, *Ophryoscolex*, and undefined genus within family Neocallimastigaceae were observed.

The associations between rumen FA and bacterial taxa were represented by a clustered image map (Figure 5) inferred from the rCCA analysis. Genera *Ruminococcus*, *Anaerovibrio*, *Butyrivibrio*, *Bulleidia*, *Methanosphaera*, *SHD231*, *Mogibacterium*, and *Methanobrevibacter* and undefined genera within the families Veillonellaceae and Coriobacteriaceae were positively correlated with the total MUFA and some BH intermediates (C18:1 trans4, C18:1 trans-5, C18:1 trans 6-7-8, C18:1 trans-9, C18:1 trans-10, C18:1 trans-12, C18:1 trans13-14, C18:1 trans-15, C18:1 trans-16, C18:1 cis-11, C18:1 cis-9, C18:2 trans-11 cis-15, and C22:1 cis13) while these FA were negatively correlated with the genera *Clostridium*, *Succiniclasticum*, *Lachnospira*, *Blautia*, *Pyramidobacter*, *Pseudobutyrivibrio*, and *Coprococcus*; undefined genera within order Clostridiales and RF39; and undefined genera within families S247, Ruminococcaceae, and Lacnospiraceae.

Genera *Clostridium, Succiniclasticum, Lachnospira*, *Blautia, Pyramidobacter*, *Pseudobutyrivibrio*, and *Coprococcus*; undefined genera within orders Clostridiales and RF39; and undefined genera within families S247, Ruminococcaceae, and Lacnospiraceae were positively correlated with total SFA, concretely with C12:0, C14:0, C16:0, and 13-oxo C18:0, while these FA were negatively correlated with genera *Ruminococcus* and *Anaerovibrio* and undefined genera within family Veillonellaceae.

Genera *Prevotella*, *Treponema*, *YRC22*, *CF231*, *Ruminococcus*, and *Anaerovibrio* and undefined genera within families Succinivibrionaceae, Paraprevotellaceae, and S247 were positively correlated with long chained saturated fatty acids, concretely with C17:0, C18:0, C19:0, C20:0, and C22:0, while these FA were negatively correlated with genera p-75-a75, *Butyrivibrio*, *Clostridium*, *Coprococcus*, *Blautia*, and *Shuttleworthia*; undefined genera within families Chistensenellaceae, RF16, Ruminococcaceae, and Lacnospiraceae; and undefined genera within orders RF39 and Clostridiales.

### 3.3. Milk Fatty Acid Composition

The proportions of most short and medium chain SFA were not modified by the dietary treatments (Table 7), except for C13:0, which was increased in the milk fat of CPRC-fed cows (*p* = 0.043).

The diet with CPRC did not modify the milk proportions of C18:1 cis-9 (*p* = 0.628), C18:1 cis-11 (*p* = 0.427), or C18:1 trans-11 (*p* = 0.650) but increased C18:1 trans-6 (*p* = 0.001), C18:1 trans-10 (*p* = 0.034), and C18:1 trans-12 (*p* = 0.043). The total MUFA (*p* = 0.495), cis MUFA (*p* = 0.633) and trans MUFA (*p* = 0.062) were not modified when the CPRC diet was fed.

As shown in Table 8, feeding a diet with CPRC did not affect the total PUFA in milk (*p* = 0.625). The use of CPRC did not affect milk fat C18:2 cis-9 trans-11 CLA (*p* = 0.834) or total CLA (*p* = 0.711) but reduced C18:3n-6 (*p* = 0.043) and increased C18:3n-3 (*p* = 0.008) and C20:1n-9 cis-11 (*p* < 0.001) proportions. The milk ratio of PUFA:SFA did not differ between treatments (*p* = 0.507), whereas the n6:n3 ratio was lower in the CPRC experimental group (*p* = 0.028).

### 3.4. Milk Yield and Milk Composition

The milk composition in terms of crude fat (*p* = 0.100), crude protein (*p* = 0.203), or lactose (*p* = 0.556) proportions did not differ in the experimental group fed a diet with CPRC compared with the control group (Table 9). Similarly, feeding a diet with CPRC did not affect the yields of milk (*p* = 0.304), FCM (*p* = 0.679), crude fat (*p* = 0.633), crude protein (*p* = 0.616), or lactose (*p* = 0.485).

### 3.5. Pasteurized Milk Perceptibility and Sensory Properties

In the triangle test, consumers were able to differentiate between the milk of the CTR and CPRC groups (*p* < 0.001). Feeding a diet with CPRC enhanced the overall acceptability by 0.43 points out of 9 (*p* = 0.047) and by improving the flavor by 0.52 points out of 9 (*p* = 0.021). Appearance, odor, or texture were not perceived being as different (*p* > 0.05; Table 10)

## 4. Discussion

The proportion of total SFA and specifically short/medium-chain FA (C12:0, C14:0, and C16:0) in the ruminal liquid mimicked that of the diets and is in agreement with the changes observed in other in vitro studies using CPRC [9]. Moreover, feeding a concentrate with CPRC induced some relevant changes related to the ruminal FA BH process. Although the total SFA decreased in rumen fluid in diets with CPRC, the C18:0–C22:0 proportions increased. The main FAs present in the experimental concentrates were C18:1 cis-9 (23.4 vs. 41.0 g/100 g FA for CTR and CPRC, respectively) and C18:2 cis9 cis12 (35.0 vs. 32.7 g/100 g FA for CTR and CPRC, respectively). These FA were subjected to a BH process in the rumen carried out by ruminal bacteria that ended up in the formation of C18:0 [35]. The higher proportion of ruminal C18:0 found with CPRC can be due to the fact that the CPRC diet provided greater amounts of C18 UFAs compared with the CTR diet. Other authors have observed the same trends using CPRC [9] and cold-pressed sunflower cake, also rich in C18 UFAs [36].

Plant lipid sources rich in UFA have been related to an increase in the C18:1 production in the rumen [37,38]. The extent of rumen BH of C18 UFAs is known to vary between 58 and 100% [39]. However, it is important to highlight that the final reduction step of UFA to C18:0 is considered rate limiting, and therefore, C18:1 intermediates (mainly C18: trans-11) can accumulate and flow out of the rumen, mainly when excessive amounts of UFA are ingested [40,41]. Considering the higher ruminal proportions of total MUFA, especially C18:1 cis-9, and total trans-MUFA, especially C18:1 trans-11, this seemed to be the case when feeding a diet with CPRC in the present study.

Moreover, an effect of the type of fat present in the CPRC on the microorganisms involved in the BH process cannot be precluded. Huws et al. [42] proposed that uncultured bacteria belonging to genera *Anaerovorax*, *Prevotella*, *Lachnospiraceae Incertae Sedis*, *Ruminococcus*, *Butyrivibrio*, *Pseudobutyrivibrio*, *Tanerella*, unclassified Bacteroidales, Clostridia and Clostridiales, Ruminococcaceae, Lachnospiraceae, Prevotellaceae, and Porphyromonadaceae might be implicated in ruminal C18:1 trans-11 formation. Other authors also stated that other genera including genus *Ruminococcus,* as one of the most prevalent in the rumen, are involved in ruminal C18: trans-11 formation [43]. In this sense, we observed an increase in the RA of genera *Ruminococcus* and some OTUs of genus *Prevotella* with CPRC, whereas *Clostridium* and the undefined genus within family Lachnospiraceae RA decreased with CPRC. However, although C18:1 trans-11 ruminal concentrations increased with diets containing CPRC, no direct relationship of any specific bacterial genus was observed with C18:1 trans-11 in the present study. Although bacterial species involved in ruminal C18:1 trans-10 formation are not well known, some authors observed ruminal formation of this FA by *Ruminococcus albus* [43,44]. In agreement with these observations we observed an increased RA in genus *Ruminococcus* in the CPRC experimental group, and this genus presented a positive relationship with C18:1 trans-10 concentrations in the rumen contents in the clustered image map.

Regarding the last step of ruminal BH, although *Butyrivibrio proteoclasticus* is the only bacterial species known to reduce C18:1 FA to C18:0 [35,45], non-cultivated *Butyrivibrio, Pseudobutyrivibrio*, and other unknown Lachnospiraceae strains could play a role in the final BH step [46]. In the current study, only the RA of *Blautia* (family Lachnospiraceae) and undefined genera within family Lachnospiraceae were decreased in ruminal contents of cows fed a diet with CPRC. Furthermore, there was a negative correlation between these taxa and trans C18:1 intermediates, potentially suggesting that these genera were involved in ruminal 18:0 formation though minor BH pathways [47]. The RA of the order RF39 was also decreased in cows fed with CPRC and was negatively correlated with 18:1 isomers, which agrees with the results observed in [48] when supplementing a fat rich in PUFA to goats. These authors hypothesized that genera within this order might also be implicated in ruminal 18:0 formation, a hypothesis that is also supported by our results. Another alternative explanation is that feeding CPRC reduced the biohydrogenating activity of *B. proteoclasticus* instead of its RA, but to test this hypothesis, metatranscriptomic assays should be performed and are far from the objective of the present study.

The use of a diet with CPRC seemed not to affect the first steps of the BH pathway of C18:2 in the rumen, since neither the main intermediate C18:2 cis-9 trans 11 CLA proportion nor the proportions of other minor alternative intermediates proportions were altered in the rumen [43]. This may be explained by the great extent of ruminal BH that happens with linoleic acid (up to 95%; [49]). For the C18:3 BH process, none of the main intermediates of the first stages of the BH process seemed to be affected by the diet with CPRC. However, some alternative pathways seemed to be affected. Dewanckele et al. [43] showed that a minor intermediate pathway for BH of C18:3 in the rumen was the hydrogenation and isomerizarion to C18:2 cis-12,cis15 and C18:2 trans-12,cis-15 and the posterior hydrogenation to some C18:1 isomers (C18:1 cis-11, C18:1 cis-12, C18:1 trans-12, C18:1 cis-15, C18:1 trans-15, and C18:1 trans-16), which were finally hydrogenated to C18:0. We observed that some of these intermediates (C18:1 cis-11, C18:1 trans-12, C18:1 cis-15, and C18:1 trans-16) increased in the rumen of cows of the CPRC group. This would be related to an inhibitory effect of the lipids present in the CPRC on the last step of BH of these FA to C:18:0. As mentioned before, some unknown Lachnospiraceae strains might play a role in the final BH step [46]. In this sense, we observed a negative relationship of these intermediates with the RA of Blautia (family Lachnospiraceae) and the undefined genera within family Lachnospiraceae, and we also observed that the RA of these genera was decreased in the ruminal contents of cows fed a diet with CPRC.

The contribution of protozoa and fungi in the rumen to the BH process has been reported as negligible and mainly associated with activity of protozoa ingested bacteria [50,51]. However, it is recognized that rumen protozoa contain proportionally more UFA than rumen bacteria and thus could play an important role in increasing CLA or C18:1 trans-11 ruminal proportions and can contribute in a significant way to the flow of UFAs leaving the rumen [52,53,54]. This is in agreement with the increased RA of some rumen ciliates and the increased ruminal C18:1 trans-11 concentration in the ruminal content of CPRC experimental group. Conversely, some authors reported decreased ciliate protozoa when rapeseed oil was included in the diet of sheep [55]. However, the level of inclusion and the physical form of the fat supplement could play a role in the effect towards protozoa population. In addition, Newbold et al. [56] suggested that, although high dietary lipid concentration is toxic to protozoa, the antiprotozoal effect of fat depends on the FA composition, with medium chain FA being more effective in reducing ciliates than PUFA.

In the present paper, increasing the dietary UFA content with the use of CPRC in the cow ration did not have a great effect on microbial populations diversity (alpha and beta diversity indices) but led to changes in some bacterial and eukaryotic taxa. However, CRPC partially replaced other ingredients in the concentrate. In this regard, differences not only in the FA profile but also in the chemical composition of both concentrates evaluated cannot be ignored and might also contribute to explain some subtle differences in ruminal microbial populations. It was observed that these changes could modify the BH process in the rumen. However, changes observed in ruminal FA profile had a slight reflect on milk FA profile. Opposite to those changes observed in rumen contents, a diet with CPRC did not reduce proportions of total SFA in milk, probably due to a compensation of the observed lower short chain FA in the rumen with de novo synthesis of these FA in the mammary gland. This is in agreement with the results observed by other authors on sheep milk [6] but differs from the results observed by [5] with sheep and by [57] with dairy cows and with the idea that including long chain UFA in the ration decreases milk short and medium chain FA through inhibition of de novo synthesis in the mammary tissue [58,59]. The abundance of C18:1 cis-9 and PUFA in plant lipids is known to alter the distribution of trans FA in milk fat [60], and in agreement with our results, supplementation with canola or rapeseed has been previously related to the increases in milk trans FA concentrations [57,61]. However, other authors have observed no changes [58,62]. This inconsistency could be partially explained by the physical form of the fat supplement. Givens et al. [58] observed that the physical properties of the rapeseed supplement were crucial to observing important changes in the milk FA profile. While rapeseed oil or rapeseed milled increased milk C18:1 isomers (cis and trans), diets containing whole rapeseeds resulted in minor changes, highlighting the key role of the bioavailability of lipids.

Although we observed an increase in ruminal proportions of C18:1 trans-11, which is known as the main precursor of C18:2 cis-9 trans-11 CLA synthesis in mammary tissue, the proportion of these CLA isomer was not increased in the milk of the CPRC group. Pascual et al. [6] also observed no effect of feeding CPRC on milk C18:2 cis-9 trans-11 CLA proportions, but these authors observed a clear increase in the milk C18:1 trans-11 proportions. Moreover, this is in disagreement with previous studies where rapeseed-based feeds increased the milk proportions of C18:2 cis-9 trans-11 CLA and other CLA isomers [57,58,62]. Regarding n3 FA, our results agree with previous studies that have pointed out that supplementing with CPRC, rich in C18:3-n3, increases milk long-chain *n*-3 FAs [5,6].

Finally, regarding production performance, no detrimental effects of using CPRC as a UFA rich lipid source in dairy cow rations was observed, which is consistent with other studies with dairy sheep [5,6] or beef cattle [4]. Moreover, the changes observed in the milk FA profile did not affect the milk sensory quality in a negative manner. In dairy rations, one key factor for the practical use of new feedstuffs, especially those rich in lipids, is to ensure that the final product’s taste remains pleasant and free of off-flavors. In the present study, as mentioned, not only was there no negative effect but a better flavor and overall acceptability was observed for milk from CPRC-fed cows compared with the control. Flavor is known as one of the key attributes for acceptability, and among the variables affecting milk flavor, fat is pointed out as one of the most important ones [63], so even slight changes observed in the milk FA profile seemed to be enough to affect milk flavor in a positive way. Other authors have observed no effect of feeding CPRC on sheep curd [6] or cheese [7] sensory properties, whereas the authors of [64] observed similar results in dairy cattle milk when feeding cold-pressed sunflower cake rich in UFA.

This study provided a new insight into the effects of using CPRC as an alternative lipid supplement in dairy cows’ diets on ruminal BH of dietary FA and ruminal microbial communities and how the changes exerted in the rumen influence productive performance, milk FA profile, and milk sensorial quality.

## 5. Conclusions

In conclusion, a diet with CPRC affected some microbial taxa at the rumen level, modified the fatty acid biohydrogenation process, and resulted in a slight improvement in the milk fatty acid profile and consumer acceptance without detrimental effects on milk production and quality.

## Figures and Tables

**Figure 1 animals-11-02553-f001:**
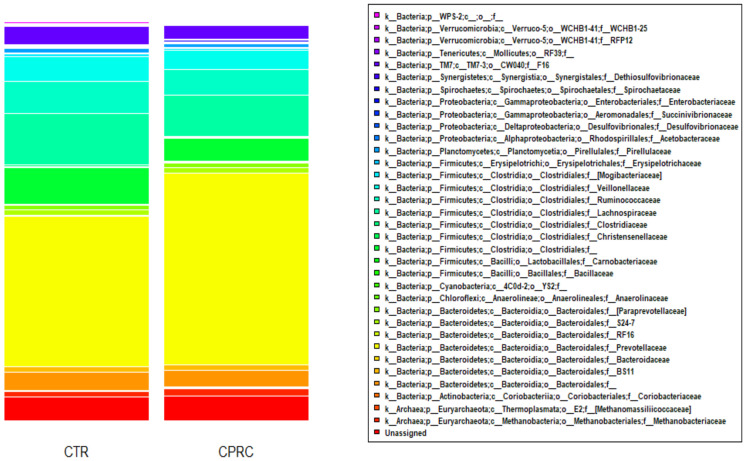
Bacterial community composition at the family level in the ruminal samples of cows (*n* = 18) when fed a control concentrate (CTR) and a concentrate with cold-pressed rapeseed cake (CPRC).

**Figure 2 animals-11-02553-f002:**
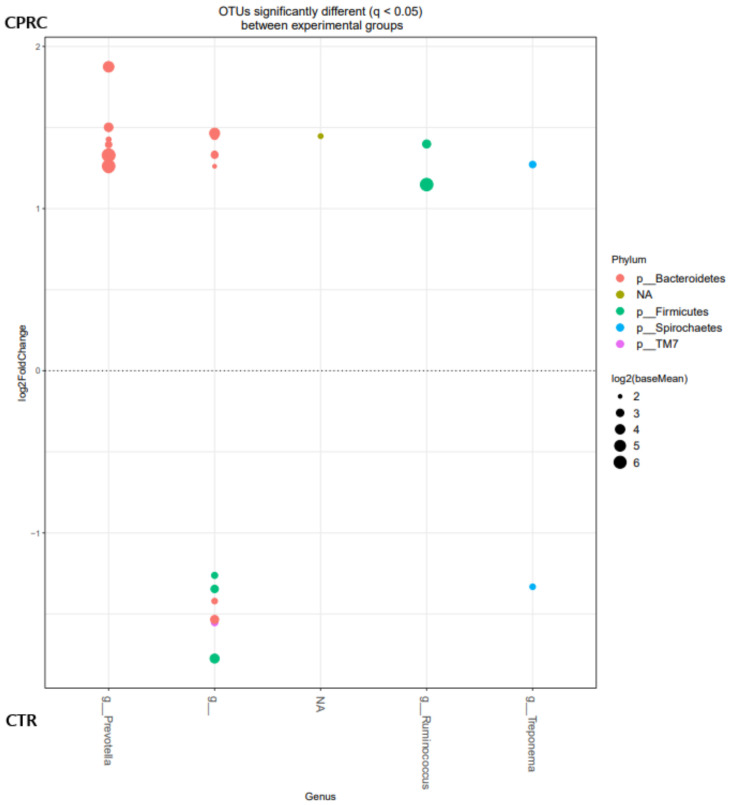
OTUs at the genus level were significantly different (q < 0.05) between rumen samples of cows fed the control (CTR; below) and concentrate with cold-pressed rapeseed cake (CPRC; above). Each point represents a single OTU colored by phylum and grouped on the *x*-axis by taxonomy. The size of the point reflects the log2 mean abundance of the sequence data.

**Figure 3 animals-11-02553-f003:**
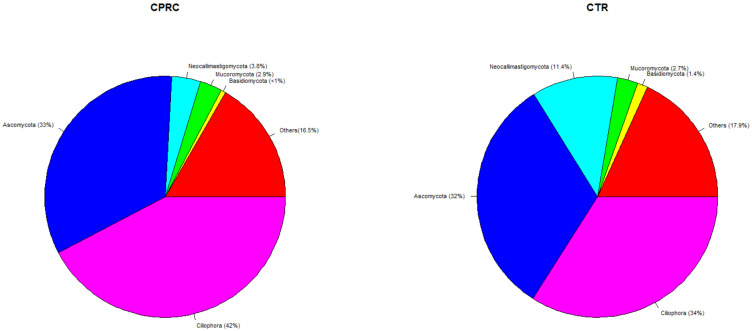
Mean relative abundances of Eukaryote phyla in the rumen samples of cows fed a concentrate with cold-pressed rapeseed cake (CPRC) or a control concentrate (CTR). “Others” include phyla with relative abundance <0.1% and phyla related to feed and the host (Charophyta, Chlorophyta, and Chordata).

**Figure 4 animals-11-02553-f004:**
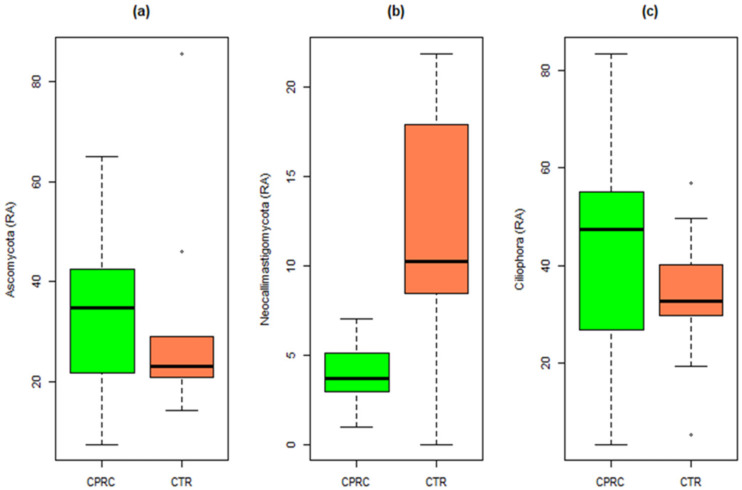
Relative abundances of the three main Eukaryote phyla in rumen samples of cows fed a concentrate with cold-pressed rapeseed cake (CPRC) or a control concentrate (CTR). (**a**) Relative abundance of Ascomycota, (**b**) relative abundance of Neocallimastigomycota, and (**c**) relative abundances of Ciliophora.

**Figure 5 animals-11-02553-f005:**
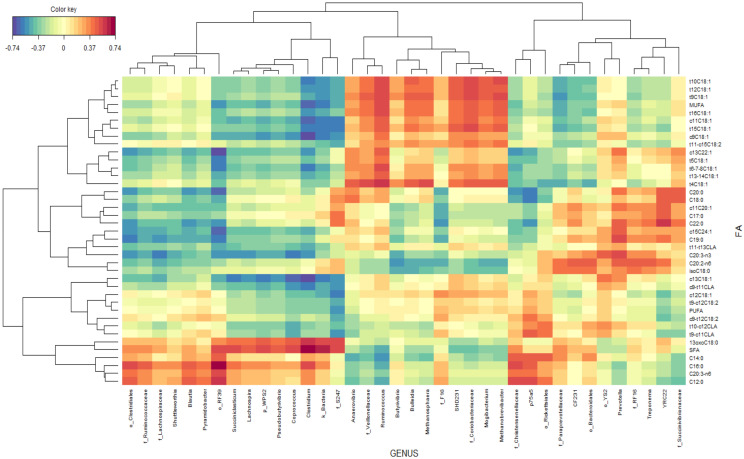
Associations between bacterial genus and ruminal fatty acid proportions independent of the experimental group. Clustered image map based on the regularized canonical correlations between bacterial relative abundances at the genus level and ruminal proportions of fatty acids. Significative correlations are colored following the key shown.

**Table 1 animals-11-02553-t001:** Ingredients and chemical composition of basal forage ration and experimental concentrates (CTR, CPRC).

Item	Experimental Concentrates		
	CTR	CPRC	Basal Forage
Ingredients composition (g/kg DM)			
Corn	225	156	
Soybean meal	210	148	
Cold-pressed rapeseed cake	0	234	
Palm Kernel meal	167	0	
Distillers dried grains	139	18	
Barley	108	236	
Wheat	55	150	
Molasses	20	20	
Hydrogenated palm fat	20	0	
Alfalfa pellets	20	0	
Minerals and vitamins ^1^	36	38	
Maize silage			295
Grass silage			615
Barley straw			90
Chemical composition (g/kg DM)			
Dry matter	883	885	467
Starch	290	310	61
Crude protein	190	190	121
Neutral detergent fibre	205	185	405
Acid detergent fibre	90	92	315
Acid detergent lignin	23	20	41
Ether extract	60	63	22
Net energy content (UFL)	1	1	

CTR: control; CPRC: cold-pressed rapeseed cake; DM: dry matter; UFL: calculated net energy requirements for lactation equivalent of 1 kg standard air-dried barley. ^1^ Contained (g/kg) calcium (270), magnesium (60), sodium (40), phosphorus (40) zinc (5.0), manganese (4.0), copper (1.5); (mg/kg), iodine (500), cobalt (50), selenium (15); (IU/g) retinyl acetate (500), cholecalciferol (100), and DL-α-tocopheryl acetate (0.5).

**Table 2 animals-11-02553-t002:** Fatty acid composition of experimental concentrates.

Key Fatty Acids (g/100 g of Total Fatty Acids)	CTR	CPRC
C12:0	7.61	0.090
C13:0	0.030	0.010
C14:0	3.19	0.280
C150	0.040	0.100
C16:0	23.7	10.4
C17:0	0.080	0.070
C18:0	2.90	1.82
C20:0	0.230	0.470
C22:0	0.150	0.310
C23:0	0.060	0.060
C24:0	0.270	0.240
C16:1 cis-9	0.120	0.270
C18:1 cis-9	23.4	41.0
C18:1 cis-11	1.13	4.47
C20:1 cis-11	0.220	0.840
C18:2 cis-9 cis-12	35.0	32.7
C18:3 cis-9 cis-12 cis-15	1.63	6.30

CTR: control; CPRC: cold-pressed rapeseed cake.

**Table 3 animals-11-02553-t003:** Effect of including cold-pressed rapeseed cake in the concentrate of dairy cows on ruminal saturated fatty acid composition (LSM, *n* = 18).

Item (g/100 g FA)	CTR	CPRC	SED	*p*-Value
C12:0	1.99	0.33	0.144	<0.001
C13:0	0.042	0.087	0.0225	0.208
C13:0 iso	0.041	0.044	0.0056	0.785
C14:0	2.94	2.07	0.195	0.019
C15:0	0.728	0.699	0.0482	0.554
C15:0 iso	1.05	1.08	0.195	0.899
C15:0 anteiso	0.258	0.220	0.0422	0.445
C16:0	19.7	13.6	0.25	<0.001
C16:0 iso	0.185	0.177	0.0276	0.839
C17:0	0.360	0.438	0.0180	0.005
C17:0 iso	0.200	0.215	0.0150	0.494
C17:0 anteiso	0.231	0.225	0.0255	0.874
C18:0	48.5	52.6	1.08	0.007
C18:0 iso	0.072	0.072	0.0045	0.980
10-oxo-C18:0	0.433	0.490	0.0576	0.436
13-oxo-C18:0	0.213	0.206	0.0207	0.790
C19:0	0.068	0.078	0.0024	0.020
C20:0	0.729	0.943	0.0222	<0.001
C22:0	0.486	0.595	0.0180	<0.001
C23:0	0.136	0.145	0.0092	0.543
C24:0	0.581	0.571	0.0200	0.679
∑ SFA	79.5	75.6	0.59	<0.001

CTR: control, CPRC: cold-pressed rapeseed cake, SED: standard error of the difference, FA: fatty acid, SFA: saturated fatty acid.

**Table 4 animals-11-02553-t004:** Effect of including cold-pressed rapeseed cake in the concentrate of dairy cows on ruminal unsaturated fatty acid composition (LSM, *n* = 18).

Item (g/100 g FA)	CTR	CPRC	SED	*p*-Value
C14:1 trans-9	0.056	0.071	0.0096	0.314
C16:1 cis-7	0.104	0.111	0.0126	0.699
C16:1 cis-9	0.054	0.061	0.0055	0.403
C16:1 trans-9	0.005	0.006	0.0006	0.287
C18:1 cis-9	3.66	4.48	0.204	0.028
C18:1 cis-11	0.491	0.671	0.0371	0.003
C18:1 cis-12	0.805	0.778	0.0429	0.609
C18:1 cis-13	0.132	0.143	0.0081	0.165
C18:1 cis-15	0.206	0.233	0.0113	0.096
C18:1 cis-16	0.149	0.159	0.0068	0.341
C18:1 trans-4	0.198	0.259	0.0115	0.010
C18:1 trans-5	0.099	0.141	0.0041	<0.001
C18:1 trans-6-7-8	0.623	0.889	0.0243	<0.001
C18:1 trans-9	0.416	0.530	0.0346	0.058
C18:1 trans-10	0.797	0.972	0.0571	0.054
C18:1 trans-11	2.68	3.55	0.140	0.005
C18:1 trans-12	0.91	1.03	0.033	0.035
C18:1 trans-13-14	1.49	1.80	0.070	0.020
C18:1 trans-15	1.03	1.13	0.057	0.265
C18:1 trans-16	0.763	0.855	0.0305	0.078
C20:1 cis-11	0.021	0.035	0.0035	0.027
C22:1 cis-13	0.021	0.047	0.0013	<0.001
C24:1 cis-15	0.087	0.130	0.0051	<0.001
C18:2 cis-9 cis-12	1.70	1.47	0.119	0.223
C18:2 cis-9 trans-12	0.021	0.018	0.0022	0.012
C18:2 trans-11 cis-15	0.259	0.311	0.0303	0.260
C18:2 trans-9 trans-12	0.013	0.015	0.0015	0.441
C18:2 cis-9 trans-11 CLA	1.42	1.66	0.167	0.354
C18:2 trans-9 cis-11 CLA	0.055	0.047	0.0062	0.146
C18:2 trans-10 cis-12 CLA	0.066	0.061	0.0067	0.523
C18:2 trans-11 trans-13 CLA	0.039	0.050	0.0042	0.113
C18:3n-3	0.389	0.453	0.0296	0.150
C20:2n-3	0.003	0.004	0.0006	0.307
C20:2n-6	0.019	0.022	0.0022	0.426
C20:3n-6	0.008	0.005	0.0013	0.293
C20:4n-6	0.016	0.011	0.0043	0.148
∑ MUFA trans	9.1	11.2	0.30	0.002
∑ MUFA cis	5.73	6.85	0.284	0.028
∑ MUFA	14.8	18.1	0.45	0.002
∑ PUFA	4.47	4.37	0.324	0.829
∑CLA	1.86	2.09	0.189	0.430
∑ *n*-3	0.411	0.472	0.0305	0.157
∑ *n*-6	0.063	0.061	0.0078	0.803
*n*-6:*n*-3	0.154	0.128	0.0154	0.263
C18:1 trans-10:trans-11	0.306	0.278	0.0235	0.377

CTR: control, CPRC: cold-pressed rapeseed cake, SED: standard error of the difference, FA: fatty acid, MUFA: mono-unsaturated FA, PUFA: poly-unsaturated FA, CLA: conjugated linoleic acid.

**Table 5 animals-11-02553-t005:** Effect feeding cold-pressed rapeseed on ruminal microbiota diversity measurements in lactating cows.

Diversity Indices	Treatment		SEM	*p*-Value
16S rRNA	CTR	CPRC		
Observed OTU	10,126	10,684	290.7	0.194
Chao1	18,641	18,994	334.5	0.466
Phylogenetic diversity	380	390	6.1	0.286
Shannon	9.72	9.44	0.153	0.207
Coverage (%)	94.2	94.8	0.20	0.051
18S rRNA				
Observed OTU	1336	1180	77.0	0.172
Chao1	2453	2446	137.7	0.971
Phylogenetic diversity	19.7	19.0	0.90	0.609
Shannon	5.29	5.39	0.157	0.661
Coverage (%)	99.4	99.3	0.04	0.215

CTR: control; CPRC: cold-pressed rapeseed cake; SEM: standard error of the mean.

**Table 6 animals-11-02553-t006:** Relative abundances (% of total ciliphora sequences) of Protozoal genera in the rumen digesta samples of cows fed a concentrate with cold-pressed rapeseed cake or a control concentrate.

Genus	Treatment		SEM	*p*-Value
	CTR	CPRC		
*Entodinium*	15.0	23.9	2.76	0.039
*Diplodinium*	3.2	5.7	2.22	0.442
*Dasytricha*	6.6	5.3	1.75	0.615
*Isotricha*	12.6	7.6	2.86	0.232
*Ophryoscolex*	2.6	2.1	0.89	0.675
*Platyophryida*	0.03	1.34	0.808	0.269
*Eremoplastron*	0.003	0.018	0.0079	0.195
*Eudiplodinium*	0.0003	0.0002	0.00018	0.683
*Polyplastron*	0.005	0.008	0.0042	0.596
udG_Trichostomatia	56.9	52.9	4.37	0.521
udG_Haptoria	2.9	0.4	1.30	0.204
udG_Litostomatea	0.012	0.012	0.0026	0.844
Other groups ^a^	0.130	0.820	0.4510	0.295

CTR: control concentrate; CPRC: concentrate with cold-pressed rapeseed cake; SEM: standard error of the mean; ^a^: relative abundance < 0.5% in any one sample; udG: undefined genera.

**Table 7 animals-11-02553-t007:** Effect of including cold-pressed rapeseed cake in the concentrate of dairy cows on milk saturated fatty acids composition (LSM, *n* = 18).

	CTR	CPRC	SED	*p*-Value
FA (g/100 g FA)				
C4:0	3.27	3.11	0.137	0.273
C6:0	1.80	1.86	0.059	0.353
C8:0	0.695	0.814	0.0560	0.143
C10:0	2.34	2.60	0.138	0.233
C11:0	0.034	0.041	0.0026	0.077
C12:0	3.46	3.25	0.171	0.427
C13:0	0.061	0.073	0.0033	0.043
C14:0 iso	0.096	0.081	0.0064	0.057
C14:0	12.2	12.0	0.37	0.734
C15:0 anteiso	0.210	0.205	0.0092	0.610
C15:0	0.881	0.924	0.0469	0.520
C16:0	31.0	28.7	1.08	0.192
C17:0	0.364	0.391	0.0080	0.057
C18:0	10.8	11.8	0.55	0.251
C20:0	0.146	0.163	0.0069	0.133
C21:0	0.038	0.037	0.0017	0.734
C23:0	0.030	0.029	0.0017	0.560
C24:0	0.052	0.049	0.0030	0.451
∑BCFA	0.783	0.742	0.0282	0.276
∑SFA	70.9	69.6	1.05	0.413
De Novo	42.1	41.1	0.88	0.443

FA: Fatty acids; BCFA: Branched-chain fatty acids; SFA: saturated fatty acid; CTR: control; CPRC: cold-pressed rapeseed cake; SED: standard error of the difference.

**Table 8 animals-11-02553-t008:** Effect of including cold-pressed rapeseed cake in the concentrate of dairy cows on milk unsaturated fatty acids composition (LSM, *n* = 18).

	CTR	CPRC	SED	*p*-Value
FA (g/100 g FA)				
C18:1 cis-9	22.7	23.4	0.96	0.628
C18:1 cis-11	0.265	0.256	0.0092	0.427
C18:1 trans-6	0.601	0.772	0.0366	0.001
C18:1 trans-10	0.386	0.511	0.0362	0.034
C18:1 trans-11	1.00	1.05	0.078	0.650
C18:1 trans-12	0.394	0.456	0.0200	0.043
C18:2 trans-9 trans-12	0.049	0.043	0.0061	0.536
C18:2 cis-9 cis-12	1.84	1.88	0.106	0.782
C18:2 cis-9 trans 11 CLA	0.666	0.676	0.0451	0.834
C18:3n-6	0.025	0.021	0.0013	0.043
C18:3n-3	0.262	0.326	0.0123	0.008
C20:1n-9 cis-11	0.044	0.062	0.0019	<0.001
C20:2n-6	0.021	0.022	0.0007	0.341
C20:3n-6	0.078	0.070	0.0037	0.155
C20:4n-6	0.024	0.024	0.0015	0.952
C20:3n-3	0.113	0.110	0.0057	0.394
C22:2n-6	0.037	0.035	0.0017	0.341
C22:6n-3	0.051	0.050	0.0026	0.537
C24:1n-9	0.011	0.012	0.0007	0.239
∑ cis MUFA	23.0	23.7	0.96	0.633
∑ trans MUFA	2.38	2.79	0.154	0.062
∑MUFA	28.0	29.1	1.00	0.495
∑PUFA	3.43	3.53	0.142	0.625
∑CLA	0.901	0.928	0.0482	0.711
C18:1 trans-10/trans-11	0.394	0.487	0.0261	0.045
*n*-6:*n*-3	4.68	4.24	0.173	0.028
PUFA:SFA	0.049	0.051	0.0024	0.507

FA: fatty acids; MUFA: monounsaturated fatty acids; PUFA: polyunsaturated fatty acids; CLA: conjugated linoleic acid. SFA: saturated fatty acids; CTR: control; CPRC: cold-pressed rapeseed cake; SED: standard error of the difference.

**Table 9 animals-11-02553-t009:** Effect of feeding cold-pressed rapeseed cake on milk yield and composition of lactating dairy cows (LSM, *n* = 18).

	Treatments		SED	*p*-Value	
	CTR	CPRC		Week	Treatment
Milk yield, kg/d	23.8	25.2	0.97	<0.001	0.304
Milk fat, %	4.00	4.35	0.175	0.043	0.100
Yield, kg/d	0.946	0.954	0.0614	0.004	0.633
3.5% FCM, kg/d	25.7	28.7	1.90	<0.001	0.179
Milk crude protein, %	3.30	3.42	0.064	<0.001	0.203
Yield, kg/d	0.752	0.790	0.0541	0.003	0.616
Milk lactose, %	4.89	4.86	0.035	0.609	0.556
Yield, kg/d	1.09	1.19	0.108	0.002	0.485

FCM: fat corrected milk; CPRC: cold-pressed rapeseed cake; CTR: control; SED: standard error of the difference.

**Table 10 animals-11-02553-t010:** Effect of feeding cold-pressed rapeseed cake on milk sensorial quality of lactating dairy cows (*n* = 40).

	CTR	CPRC	SED	*p*-Value
Overall acceptability	5.83	6.26	1.646	0.047
Appearance	6.67	6.80	1.414	0.494
Odour	5.86	5.90	1.403	0.818
Texture	6.14	6.47	1.455	0.080
Flavour	5.47	5.99	1.756	0.021

CTR: control; CPRC: cold-pressed rapeseed cake; SED: standard error of the difference.

## Data Availability

The datasets generated and/or analyzed during the current study are available from the corresponding author upon reasonable request.

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
