# Peer review of "Evaluating the Inclusion of Cold-Pressed Rapeseed Cake in the Concentrate for Dairy Cows upon Ruminal Biohydrogenation Process, Ruminal Microbial Community and Milk Production and Acceptability"

_animals, 2021, doi:10.3390/ani11092553_

Round 1
Reviewer 1 Report
This paper evaluated the cold pressed rapeseed cake as a feedstuff for dairy cows. It is of great significance to expand feed resources and enrich dairy feed diversification. The overall writing quality of the manuscript is good. A few of comments as follows:
- The title of the article is probably not appropriate enough. The processing technology of cold pressed rapeseed cake is green, but the main content of the manuscript is the influence of rapeseed cake on dairy production, which does not fully reflect the green characteristics. In fact, there are many studies on the application of rapeseed cake in dairy diet. But from the content of the manuscript, I can see that the author mainly focused on the fatty acids of feed, rumen content, and milk, and the rumen microbial hydrogenation. It is suggested that the author optimize the title from this point.
- As for the dietary formula of the manuscript, it can be seen that the author not only used rapeseed cake to replace palm kernel meal and soybean meal, but also adjusted distillers dried grains and other ingredients greatly. So, the problem is how to confirm that the main influencing factor of the experiment is rapeseed meal, not others. Besides, I suggest that the total amount of ingredients for the concentrates and basal diets, the dry matter, energy value, calcium and phosphorus should be added in table 1.
- As for the number of experimental animals, to be honest, it is a little small, especially including two species. The author did not show the impact of species, which I think is also an important factor for the milk yield and quality.
L244 the mixed model equation was suggested.
Author Response
Reviewer 1
This paper evaluated the cold pressed rapeseed cake as a feedstuff for dairy cows. It is of great significance to expand feed resources and enrich dairy feed diversification. The overall writing quality of the manuscript is good. A few of comments as follows:
The title of the article is probably not appropriate enough. The processing technology of cold pressed rapeseed cake is green, but the main content of the manuscript is the influence of rapeseed cake on dairy production, which does not fully reflect the green characteristics. In fact, there are many studies on the application of rapeseed cake in dairy diet. But from the content of the manuscript, I can see that the author mainly focused on the fatty acids of feed, rumen content, and milk, and the rumen microbial hydrogenation. It is suggested that the author optimize the title from this point.
The title has been changed as suggested to Effect of Feeding Cold-Pressed Rapeseed Cake on Ruminal biohydrogenation process, ruminal microbial Community and milk production and acceptability in Dairy Cows
As for the dietary formula of the manuscript, it can be seen that the author not only used rapeseed cake to replace palm kernel meal and soybean meal, but also adjusted distillers dried grains and other ingredients greatly. So, the problem is how to confirm that the main influencing factor of the experiment is rapeseed meal, not others. Besides, I suggest that the total amount of ingredients for the concentrates and basal diets, the dry matter, energy value, calcium and phosphorus should be added in table 1.
All of the ingredients were already in Table 1. Calcium and phosphorous were included as Minerals and vitamins and the total amount are shown in the footnote of the Table 1.
We have included Dry matter and concentrate net energy contents in the revised version.
As for the number of experimental animals, to be honest, it is a little small, especially including two species. The author did not show the impact of species, which I think is also an important factor for the milk yield and quality.
We agree with the reviewer that a greater number of Animals would be much better. However, we had to face limitations in terms of availability of Animals and welfare/ethics issues. Anyway, there are plenty of other manuscripts in the literature with similar and even lower number of experimental Animals. Regarding the impact of the breed we paired aimals, among other factors, on the basis of breed and randomly distributed the treatment in the pair in such a way that the variability related to the breed would be evenly distributed between treatments.
L244 the mixed model equation was suggested.
We included the equation as suggested.

Reviewer 2 Report
Goiri et al.
Animals 10.3390
Assessment of Cold-Pressed Rapeseed Cake as a “Green” Feedstuff for Dairy Cows.
Major (General or Overall) Comments to the Authors.
The authors set out to evaluate the effects of rapeseed cake on milk production outcomes and rumen microbiome. Overall, the document is reasonably well written in terms of English quality. However this reiviewer is concnerned about the use of vague and imprecise language and erroneous presentation of implications and conclusions. In this study specific objectives are not clearly defined nor exact outcomes are specified. Similarly, no clear hypothesis is presented. Specifically, You mention that you are evaluating this rapeseed cake as an alternative to other sources and then make it sounds as if this is your hypothesis. Claiming that something could be an alternative to something else does not constitute an objective/easily demonstrable or falsifiable hypothesis; this is because this claim is vague. Please consider revising the manuscript to improve clarity of your objectives, measures, and specific hypothesis. What did you expect to find on what specific measures, and did you or did you not find support for you specific hypothesis after analyzing your data?
Although from the standpoint of nutritional management the study looks at an important practical question, the authors appear to be promoting the study as a novel and “green” alternative. As for the point of novelty, much work on this has already been carried out by other research groups over the past few decades. For this reason, it is hard to see the novelty in this study, despite the authors’ claim. See for example the following meta-analysis and study example after a quick online search:
https://cdnsciencepub.com/doi/full/10.4141/cjas2011-029
https://www.sciencedirect.com/science/article/pii/S0022030215004166
As for the point about it being a green alternative: Referring to rapeseed meal as a green alternative to other protein sources is more a statement of opinion than a statement of fact. Rapeseed cake/canola meal is a very common feedstuff that has been fed to ruminants for decades. Presenting its use as if it was something “green” is very vague. How is “green” defined? Are you measuring methane output or something similar? How is rapeseed cake “green”? Are soybeans not “green”?. This particular combination of the wording including “green” along reads as a selling point for a non-scientific audience and lack objective rigor in the sense that the rapeseed cake is merely one more alternative for feeding cattle and portraying it as something else seems biased and it is not supported properly by the authors.
Finally, you need to be careful with your wording. You did not assess the effects of rapeseed cake specifically. Because you made other changes to the control diets to balance for nutrient composition, it is only fair to say that you compared “diets enriched in rapeseed vs. diets containing other ingredients”. You cannot unequivocally know the effect of rapeseed cake; you can only compare diets with or without it.
Specific comments by line.
Title: Your title needs to be specific to indicate what your study found out or what it aimed to do. In this sense the title “Assessment of Cold-Pressed Rapeseed Cake as a “Green” Feedstuff for Dairy Cows” does not appear to reflect what you actually did in your study, which was to evaluate the effect of rapeseed cake in the diet of dairy cows in comparison to diets not containing this ingredient. Please revise title accordingly. You need to objectively describe what you did or what you study evidence refraining from introducing bias or opinions. The recurrent use of the word green to refer to rapeseed does not appear as objective, as the word “green” itself seems to refer more to a societal jargon trend, that to an objective definition of what green foods are. If you intent to make this much emphasis on “green” then this needs to be clearly and objectively defined. How is a common feedstuff green? Does this mean that the feedstuffs you are replacing (e.g., soybean) are not “green”?
Ln 13: what is “green chemistry”? Your writing seems to be often reliant on this type of claims. In order to remain of objective and clear for the readers, please make sure you use specific language in your manuscript and define these types of words whenever brought up. Green chemistry sounds good, but it is not clear what it is meant by this term; therefore, its use might not be appropriate. Please be careful when you do this and revise your manuscript accordingly.
Ln 19-20: Please revise this statement: “Feeding CPRC has the advantage to slightly improve milk fatty acid profile and therefore consumer acceptance.”. What specifically is being improved in terms of fatty acid profiles that will lead consumer to accept milks derived from cows fed rapeseed cakes better? Are you referring to milk fat content of saturated and unsaturated fatty acids? If you consider that for example increased PUFA is a good thing for milk FA profile, you will need to provide evidence for this sort of claim. Current statement is vague and sounds more like an opinion. If your statement is relying solely on consumer perception, then this is highly subjective as the public is not necessarily well informed of the literature and the debate on animal fats.
Ln 26-27: your experimental design and treatments need to be described clearly. What type of cows, what stage of lactation? What was the duration of feeding treatments? How many animals per treatment? What was the parity number? Were animals randomized to treatments? Was this a longitudinal study or a Latin Square design? What were the main outcomes of interest in this study? All of this need to be clear, even if presented succinctly in the abstract.
Ln 27-42: results in abstract seem disorganized and it is currently hard to follow a sequence. You present milk FA first, then go to discuss what could be genus of rumen bacteria, etc., then go back to milk yield and FCM. This section needs to be restructure and organized.
Ln 56-57: Do you mean palm oil in humans or palm oil products fed to cows? If you are talking about effects on human health, please see references below. Current data does not support the history fear of animal fats based on SFA content. If you are referring to their effects on milk fat composition, then you need to provide evidence that feeding SFA does indeed change milk FA profile in a meaningful/significant way. Available evidence seems to indicate milk FA content of SFA changes very little when palm oil supplements are fed to cows (for reference see Adam Lock’s research at Michigan state University). Otherwise these statements appear uninformed, and vague. Please substantiate your claims, particularly when the implication that animal products can impact human health is given to readers.
Please see the following to provide context to your claims:
https://academic.oup.com/ajcn/article/91/3/535/4597110?login=true
https://pubmed.ncbi.nlm.nih.gov/24723079/
Ln 87: need citation. I recommend you take a look at the research form Wallace’s group in the UK: https://pubmed.ncbi.nlm.nih.gov/20167098/
Ln 92-94: This hypothesis is vague. How will feeding CPRC improve milk FA profile and in which way will it modify bacterial communities? What communities of rumen bacteria, which milk FA? Why? Please revise accordingly.
Ln 95-96: as mentioned above. The claim that providing more PUFA is a net good is questionable. Perhaps a good case can be made for the omega-3 portion of the rapeseed meal, but the overall idea that SFA in cow’s milk is a bad thing for human health is currently not well supported by the best available data we have.
Ln 109-110: why did your experimental periods last 8 weeks? Any particular reason for the long follow up?
Ln 134-136: Why only one day out of the 14 d you had in the covariate period? One could argue taking at least the last 2 days would be necessary given day to day variation. Similarly, on wk 2, 4, 6, 8 and 10, how many days ‘worth of samples di you take during those weeks? Only 1 day in each case? Please explain.
Ln 249-250: “Rumen and milk FA concentrations were analyzed using the previous statistical model but without considering covariates or repeated measures.” Why? It seemed that you had repeated measures for milk FA measures. Please clarify.
L284-285: you need to be careful with your wording. You did not assess the effects of rapeseed cake specifically. Because you made other changes to the control diets to balance for nutrient composition, it is only fair to say that you compared “diets enriched in rapeseed vs diets containing other ingredients”. You cannot unequivocally know the effect of rapeseed cake; you can only compare diets with or without it. Please consider this as you present and discuss your results.
Table 3 and 4: What happened top the effects of time? You had a repeated measures analysis, and one would expect to see the effects over time and their interaction (or lack thereof). Why are you not presenting those P-values. For full transparency, these need to be presented.
Table 4: Please include SFA.
Table 8: Please note that the atherogenic index is an outdated measure of risk and its usefulness is questionable. For example, fatty acids considered traditionally as atherogenic can improve HDL-C levels (aka. The good cholesterol). As mentioned earlier, currently available data contradicts the presuppositions of the diet-heart hypothesis. For a general reference, take a look a this example in addition to the references presented above:
https://ecommons.cornell.edu/bitstream/handle/1813/59846/Rico%20%28manu%29.pdf?sequence=2&isAllowed=y
Table 9: A numerical difference of 1.5 kg that did not show as significant. Did you perform a sample size calculation before establishing the number of animals in each treatment? Form these results, it may seem as if your study was underpowered to detect significant differences in production outcomes, such as milk production. This might be a false negative.
Ln 592-594: This statement needs to be revised. You cannot know that these things happened; you can only speculate that they might have happened to a certain extent. This is an important distinction. My suggestion is that you write this type of sentence in a more impartial and objective manner. State what was observed and describe potential explanations in the conditional form.
Ln 597: you use “circular production systems” 3 times in your text, but this is not defined in your manuscript. Please define at first mention, and make sure it is clear to readers if and why this is important and relevant to your study. If not relevant, please remove.
Author Response
Reviewer 2
The authors set out to evaluate the effects of rapeseed cake on milk production outcomes and rumen microbiome. Overall, the document is reasonably well written in terms of English quality. However this reiviewer is concnerned about the use of vague and imprecise language and erroneous presentation of implications and conclusions.
In this study specific objectives are not clearly defined nor exact outcomes are specified. Similarly, no clear hypothesis is presented. Specifically, You mention that you are evaluating this rapeseed cake as an alternative to other sources and then make it sounds as if this is your hypothesis. Claiming that something could be an alternative to something else does not constitute an objective/easily demonstrable or falsifiable hypothesis; this is because this claim is vague. Please consider revising the manuscript to improve clarity of your objectives, measures, and specific hypothesis. What did you expect to find on what specific measures, and did you or did you not find support for you specific hypothesis after analyzing your data?
Our hypothesis has been reworded to be clearly presented. We tried to clearly explain what we expected to find.
Although from the standpoint of nutritional management the study looks at an important practical question, the authors appear to be promoting the study as a novel and “green” alternative. As for the point of novelty, much work on this has already been carried out by other research groups over the past few decades. For this reason, it is hard to see the novelty in this study, despite the authors’ claim. See for example the following meta-analysis and study example after a quick online search:
https://cdnsciencepub.com/doi/full/10.4141/cjas2011-029
https://www.sciencedirect.com/science/article/pii/S0022030215004166
The two listed references account for solvent-extracted rapeseed cakes. Regarding the meta-analysis this study took into account the evaluation of rapessed meal (conventional) as a protein source for dairy cows.
We tried to explain in the introduction section the difference between cold-pressed rapeseed cakes and conventional cakes normally solvent-extracted. For cold-pressed rapeseed cakes, because no solvents (chemicals) are used it is claimed as a “green” process. Cold-pressed rapessed cakes are rich in fat and with high oil quality but with lower protein content than the conventionally solvent-extracted cakes, normally defatted. We agree with the reviwer in that the solvent-extracted rapeseed cakes have been extensively studied in previous works, but cold-pressed rapeseed cakes have not been so extensively studied in dairy cows. In the same way the effect of oils on ruminal biohydrogenation process and microbial communities has been already studied, but the effect of oil rich cakes, as CPRC, on ruminal biohydrogenation and microbial communities has not been so in depth studied. For this reasons we consider that the present study presents some novelty and could offer new insights for the use of these types of cakes in ruminants’ rations
As for the point about it being a green alternative: Referring to rapeseed meal as a green alternative to other protein sources is more a statement of opinion than a statement of fact. Rapeseed cake/canola meal is a very common feedstuff that has been fed to ruminants for decades. Presenting its use as if it was something “green” is very vague. How is “green” defined? Are you measuring methane output or something similar? How is rapeseed cake “green”? Are soybeans not “green”?. This particular combination of the wording including “green” along reads as a selling point for a non-scientific audience and lack objective rigor in the sense that the rapeseed cake is merely one more alternative for feeding cattle and portraying it as something else seems biased and it is not supported properly by the authors.
Probably there has been a misunderstanding. We did not refer to rapeseed as a “green” alternative. We referred to the process of obtaining the rapeseed cake, as a green alternative compared to conventional solvent-extracted cakes. We would expect soybean cake to be as green as rapeseed cake if it would be processed using a cold-pressing technology
Anyway any reference to “green” alternatives or procedures has been deleted.
Finally, you need to be careful with your wording. You did not assess the effects of rapeseed cake specifically. Because you made other changes to the control diets to balance for nutrient composition, it is only fair to say that you compared “diets enriched in rapeseed vs. diets containing other ingredients”. You cannot unequivocally know the effect of rapeseed cake; you can only compare diets with or without it.
Correct. We agree with the reviewer and we tried to revise the wording in the revised version. However, as CPRC was the major source of dietary fat in CPRC group and palm in the control group, in our opinion we can say that the effects observed in biohydrogenation process and related microbial taxa could be reasonably due to this ingredient.
Specific comments by line.
Title: Your title needs to be specific to indicate what your study found out or what it aimed to do. In this sense the title “Assessment of Cold-Pressed Rapeseed Cake as a “Green” Feedstuff for Dairy Cows” does not appear to reflect what you actually did in your study, which was to evaluate the effect of rapeseed cake in the diet of dairy cows in comparison to diets not containing this ingredient. Please revise title accordingly. You need to objectively describe what you did or what you study evidence refraining from introducing bias or opinions. The recurrent use of the word green to refer to rapeseed does not appear as objective, as the word “green” itself seems to refer more to a societal jargon trend, that to an objective definition of what green foods are. If you intent to make this much emphasis on “green” then this needs to be clearly and objectively defined. How is a common feedstuff green? Does this mean that the feedstuffs you are replacing (e.g., soybean) are not “green”?
The title has been changed and the word “green” is no longer in the title
Ln 13: what is “green chemistry”? Your writing seems to be often reliant on this type of claims. In order to remain of objective and clear for the readers, please make sure you use specific language in your manuscript and define these types of words whenever brought up. Green chemistry sounds good, but it is not clear what it is meant by this term; therefore, its use might not be appropriate. Please be careful when you do this and revise your manuscript accordingly.
We used the term “green chemistry” in the introduction section and then in the following sentence we describe the extraction process and in that way explained what this term means. As it is described in Line 65-69 the cold pressed process oil extraction relies solely on the pressure, no heat or solvents are added to the paste to assist in the extraction. This is why we called it “green chemistry”. Cold presses are usually mechanically operated and often consist of a screw device that is tightened against the paste to extract the oils. Anyway, the term Green to refer to this process has been deleted and cannot be found anynore in the text.
In line 13 (simple summary) because the words are limited we have not explain it
Ln 19-20: Please revise this statement: “Feeding CPRC has the advantage to slightly improve milk fatty acid profile and therefore consumer acceptance.”. What specifically is being improved in terms of fatty acid profiles that will lead consumer to accept milks derived from cows fed rapeseed cakes better? Are you referring to milk fat content of saturated and unsaturated fatty acids? If you consider that for example increased PUFA is a good thing for milk FA profile, you will need to provide evidence for this sort of claim. Current statement is vague and sounds more like an opinion. If your statement is relying solely on consumer perception, then this is highly subjective as the public is not necessarily well informed of the literature and the debate on animal fats.
The improvement in milk fatty acid profile is related to an increase in omega 3 fatty acids. Ragarding consumer acceptance we referred to sensorial test and the observed improved flavour and overall aceptability. We have deleted the word “therefore” to make clear that there are two different aspects but not necessarily linked one to another.
Ln 26-27: your experimental design and treatments need to be described clearly. What type of cows, what stage of lactation? What was the duration of feeding treatments? How many animals per treatment? What was the parity number? Were animals randomized to treatments? Was this a longitudinal study or a Latin Square design? What were the main outcomes of interest in this study? All of this need to be clear, even if presented succinctly in the abstract.
We tried to include some sumarized information about the experimental design. The abstract has been changed to include all this information but we have had to remove other parts to fit it to the 200 words limit.
Ln 27-42: results in abstract seem disorganized and it is currently hard to follow a sequence. You present milk FA first, then go to discuss what could be genus of rumen bacteria, etc., then go back to milk yield and FCM. This section needs to be restructure and organized.
To be coherent, we presented results in the abstract as they are presented in the paper, first ruminal (not milk) FA and microbial populations, because of the relationship between them. An second, milk production, milk FA profile and milk sensorial test.
Ln 56-57: Do you mean palm oil in humans or palm oil products fed to cows? If you are talking about effects on human health, please see references below. Current data does not support the history fear of animal fats based on SFA content. If you are referring to their effects on milk fat composition, then you need to provide evidence that feeding SFA does indeed change milk FA profile in a meaningful/significant way. Available evidence seems to indicate milk FA content of SFA changes very little when palm oil supplements are fed to cows (for reference see Adam Lock’s research at Michigan state University). Otherwise these statements appear uninformed, and vague. Please substantiate your claims, particularly when the implication that animal products can impact human health is given to readers.
Please see the following to provide context to your claims:
https://academic.oup.com/ajcn/article/91/3/535/4597110?login=true
https://pubmed.ncbi.nlm.nih.gov/24723079/
Although we agree with the reviewer we would like to draw his/her attention to the fact that we stated in that sentence “there is a debate”. Anyway the sentence has been deleted to avoid misunderstandings.
Ln 87: need citation. I recommend you take a look at the research form Wallace’s group in the UK: https://pubmed.ncbi.nlm.nih.gov/20167098/
We would like to thank the reviewer for this suggestion. A reference has been included.
Ln 92-94: This hypothesis is vague. How will feeding CPRC improve milk FA profile and in which way will it modify bacterial communities? What communities of rumen bacteria, which milk FA? Why? Please revise accordingly.
We have reworded this paragraph
Ln 95-96: as mentioned above. The claim that providing more PUFA is a net good is questionable. Perhaps a good case can be made for the omega-3 portion of the rapeseed meal, but the overall idea that SFA in cow’s milk is a bad thing for human health is currently not well supported by the best available data we have.
We agree with the reviewer. We have reworded the sentence to make the point on an improved n3:n6 ratio.
Ln 109-110: why did your experimental periods last 8 weeks? Any particular reason for the long follow up?
We took more samples than those reported in the current manuscript: SCFA, urine spot sampling for microbial protein synthesis estimation. We also estimated digestibility using internal and external makers. This is why the experimental period lasts for 8 weeks. We do not report that information in this manuscript because it would be, in our opinion, excesively long.
Ln 134-136: Why only one day out of the 14 d you had in the covariate period? One could argue taking at least the last 2 days would be necessary given day to day variation. Similarly, on wk 2, 4, 6, 8 and 10, how many days ‘worth of samples di you take during those weeks? Only 1 day in each case? Please explain.
Milk production covariate was calculated on the basis of milk production during the whole covariate period. We only took one day to sample milk because this is the usual procedure followed by the official test day recording scheme. Anyway we take the point of the reviewer and we agree that it would have been better to take more samples.
Please check the response to the previous comment to see why.
Ln 249-250: “Rumen and milk FA concentrations were analyzed using the previous statistical model but without considering covariates or repeated measures.” Why? It seemed that you had repeated measures for milk FA measures. Please clarify.
As we describe in Lines149-158 samples of ruminal contents were taken at different time points during 2 days to account for differences in the circadian cicle, but to analize these samples as we said in line 155 a pool was made for each cow. So we did not consider repeated measures because we have one measurement per animal.
For milk samples as we described in Lines142-145. Additional milk samples were collected from the AMS at each milking on wk 4 and 9 for FA profile determinations (LIGAL, Mabegondo, Spain) but samples were composited by animal and day on milk production basis and were stored and preserved at −20±5ºC with azidiol (3.3 mL/L) for FA composition analysis. So we did not consider repeated measures because we have one measurement per animal.
L284-285: you need to be careful with your wording. You did not assess the effects of rapeseed cake specifically. Because you made other changes to the control diets to balance for nutrient composition, it is only fair to say that you compared “diets enriched in rapeseed vs diets containing other ingredients”. You cannot unequivocally know the effect of rapeseed cake; you can only compare diets with or without it. Please consider this as you present and discuss your results.
We have changed the wording here and throughout the text
Table 3 and 4: What happened top the effects of time? You had a repeated measures analysis, and one would expect to see the effects over time and their interaction (or lack thereof). Why are you not presenting those P-values. For full transparency, these need to be presented.
P values for the week effect have been included but in Table 9. Table 3 and 4 refer to rumen FA acids and as mentioned before these samples were bulked prior to their analysis. Therefore these samples were not analysed considering the effect of the week.
Table 4: Please include SFA.
SFA are shown in Table 3. Due to the number of data we preferred to show unsaturated and saturated fatty acids in two independent Tables (table 3 and Table 4)
Table 8: Please note that the atherogenic index is an outdated measure of risk and its usefulness is questionable. For example, fatty acids considered traditionally as atherogenic can improve HDL-C levels (aka. The good cholesterol). As mentioned earlier, currently available data contradicts the presuppositions of the diet-heart hypothesis. For a general reference, take a look a this example in addition to the references presented above:
https://ecommons.cornell.edu/bitstream/handle/1813/59846/Rico%20%28manu%29.pdf?sequence=2&isAllowed=y
We have deleted this index
Table 9: A numerical difference of 1.5 kg that did not show as significant. Did you perform a sample size calculation before establishing the number of animals in each treatment? Form these results, it may seem as if your study was underpowered to detect significant differences in production outcomes, such as milk production. This might be a false negative.
We found a 1.5 kg difference in a mean of 24.5. This is roughly a 10% increase which is in the limit of the power of our statistical analysis. Consider as well that we found a SED of 0.97. This is really the reason why we did not find significant differences.
Ln 592-594: This statement needs to be revised. You cannot know that these things happened; you can only speculate that they might have happened to a certain extent. This is an important distinction. My suggestion is that you write this type of sentence in a more impartial and objective manner. State what was observed and describe potential explanations in the conditional form.
We observed a change in some microbial taxa. We observed a modification of the FA biohydrogenation process. We found an improvement in milk FA profile. We also found an improved consumer acceptance. All this occured without detrimental effects on milk production and quality. All these are objective results reported in tables. The extent to which happened can be seen in the figures reported in tables. The implications or importance of such changes are subjective to the interpretation of each reader, but the results reported in tables are clear. We have reworded the conclusion sentence to avoid a direct relationship among these results.
Ln 597: you use “circular production systems” 3 times in your text, but this is not defined in your manuscript. Please define at first mention, and make sure it is clear to readers if and why this is important and relevant to your study. If not relevant, please remove.
Lines have been removed
Reviewer 3 Report
Simple summary:
Line (12): Place the text on lines 21-23 after "... necessary." and before "One of ...".
Line (18): Place the text on lines 19-21 after "... feedstuffs." and before "In conclusion ...".
Abstract:
Lines (24-26): In the Abstract, it is more important to write the experimental objective than the experimental hypothesis.
Line (26): Add information about the cows: weight, age, lactation, parity, etc.
Line (36): You can summarize the P values into one, P> 0.05.
Lines (40-42): You studied CPRC and not all the by-products. Change text to: The use of CPRC to replace conventional feedstuffs……
Introduction:
The introduction lacks a good description of why they chose CRPC for their study. You could summarize the long paragraphs on policy and environmental statements and use this space to talk about the importance of the CPRC, specifically the CPRC.
The writing style of your hypothesis is confusing. You must be clear and avoid extending the ideas with additional information; you also need to be concise, clear and specific, as well as in the objective.
Line (93): Looking at the diets, the inclusion of CRPC partially replaces other ingredients as well. The statement of this line is not true.
Line (95-96): Do these lines belong to the hypothesis?
Material and methods:
The topic of “materials and methods” is repetitive. You can avoid repetitive lines and describe the description you wrote in four lines in two lines.
Line (104): Add information about cows: age, parity, days in milk, etc.
Line (105): Covariated period? Describe the covariate variable used in the “Statistics” topic.
Lines (108-109): How many days for each period?
Lines (109-110): Is it the description of the days in the previous lines? If it is, you can join the information.
Line (111): Similar amounts of energy and fat? According to Table 1, the amounts of fat and energy are different. Add energy measures (e.g. metabolizable energy, NDT, or others).
Lines (113-115): Why this feeding method? Why did you limit the consumption of the CRPC in your study and leave free access for roughage?
Table 1: Basal diet? I think you meant "basal forage"
Line (140): Mixed ration of roughage? In what proportion were the concentrate and the roughage?
Line (145): Why in these times?
Line (146): 160 mm?
Lines (149-152): How much ruminal fluid was collected for each cow?
Lines (153-154): How did you freeze these large samples?
Line (162): Fat content or ether extract content?
Line (233-235): This activity did not influence the recognition of the characteristics observed by the panelists?
Results
Tables: Standardize the decimals of SED. Use two, three, or four decimal places.
Lines (277-278; 289; 299; 308-309; 336; 344-345; 373-374; 403; 428-429; 440-441): This text is not relevant, delete it.
Discussion:
The discussion has lines that belong to the topic of results. Also, you wrote many texts comparing your results with other researchers. Finally, the explanation of the biological behavior (discussion) of its results is missing.
Lines (448-460): This text is irrelevant to discuss your data. This text can best be placed in the introduction.
Lines (462-465): How does this text correlate with the paragraph introduction about biohydrogenation (lines 461-462)?
Line (472): How is this statement true? Because CRPC concentrate has lower C18:0 fatty acid content.
Conclusion:
The conclusion must be improved.
Author Response
Reviewer 3
Simple summary:
Line (12): Place the text on lines 21-23 after "... necessary." and before "One of ...".
Changed as suggested
Line (18): Place the text on lines 19-21 after "... feedstuffs." and before "In conclusion ...".
Changed as suggested
Abstract:
Lines (24-26): In the Abstract, it is more important to write the experimental objective than the experimental hypothesis.
The objective has been written instead of the hypothesis as suggested
Line (26): Add information about the cows: weight, age, lactation, parity, etc.
Information on cows has been added to the abstract
Line (36): You can summarize the P values into one, P> 0.05.
Changed as suggested
Lines (40-42): You studied CPRC and not all the by-products. Change text to: The use of CPRC to replace conventional feedstuffs……
Changed as suggested
Introduction:
The introduction lacks a good description of why they chose CRPC for their study. You could summarize the long paragraphs on policy and environmental statements and use this space to talk about the importance of the CPRC, specifically the CPRC.
The writing style of your hypothesis is confusing. You must be clear and avoid extending the ideas with additional information; you also need to be concise, clear and specific, as well as in the objective.
Hypothesis has been reworded
Line (93): Looking at the diets, the inclusion of CRPC partially replaces other ingredients as well. The statement of this line is not true.
This statement has been delete
Line (95-96): Do these lines belong to the hypothesis?
Hypothesis has been reworded
Material and methods:
The topic of “materials and methods” is repetitive. You can avoid repetitive lines and describe the description you wrote in four lines in two lines.
Line (104): Add information about cows: age, parity, days in milk, etc.
Information on days in milk was already provided in the manuscript. Information on parity has been added in the new manuscript
Line (105): Covariated period? Describe the covariate variable used in the “Statistics” topic.
Covariates were used only for milk production and milk quality. Information on the covariate variables (duration and sampling procedures) can be seen in Lines 133-134 of the initial manuscript.
Lines (108-109): How many days for each period?
As it was described in lines 112-125. There was a 2 week covariate period, followed by the experimental period (10 weeks:2 weeks for adaptation to diets and 8 weeks for measurements). We have some difficulties here in understanding what the reviewer wishes us to do.
Lines (109-110): Is it the description of the days in the previous lines? If it is, you can join the information.
We do not understand what the reviewer wants us to do so we prefer not join the sentences.
Line (111): Similar amounts of energy and fat? According to Table 1, the amounts of fat and energy are different. Add energy measures (e.g. metabolizable energy, NDT, or others).
Energy measurements of the concentrates were added. In our opinion with a difference of 3 g/kg DM out of 60 g/kg DM, we can say that the concentrates have similar amounts of fat.
Lines (113-115): Why this feeding method? Why did you limit the consumption of the CRPC in your study and leave free access for roughage?
In the usual management when an automatic milking system is used a portion of the concentrate is given in the AMS and the rest is given mixed with the roughage offered ad libitum. In our case, the treatment (CPRC) was applied in the concentrate. We do not have individual cubicules to offer the basal diet individually, so the basal diet is group-fed. Therefore, if we wanted to have true statistical units the total amount of concentrate had to be fed individually. That is why we decided to offer the total amount of concentrate calculated to satisfy animals requirements in buckets and not left ad libitum with the basal diet.
Table 1: Basal diet? I think you meant "basal forage"
Corrected
Line (140): Mixed ration of roughage? In what proportion were the concentrate and the roughage?
As explained previously. The concentrate is not fed mixed with the basal diet. It is given individually. To avoid misunderstandings the terms mixed ration has been removed.
Line (145): Why in these times?
We tried to make a composite sample representative of the whole day, taking into account that differences occur during the day (circadian cicle). In that way we can obtain samples every 6h.
Line (146): 160 mm?
It was a mistake. It was changed to 160 cm
Lines (149-152): How much ruminal fluid was collected for each cow?
In lines 151-154 we described the volume of ruminal fluid used for the different analyses (FA profile and microbial analyses). Evidently we obtain a larger quantity of ruminal content but after sampling for FA and microbial analisis, the left over liquid was discarded. We don’t see the need of including how much ruminal fluid was collected for each cow because it is irrelevant and variable. We think it is relevant to describe the sample size (volume) of the samples that will be analysed.
Lines (153-154): How did you freeze these large samples?
As it is described in the text we freezed five samples per cow. A 100 ml pool was made for each cow with 25 mL of the liquid fraction of each ruminal extraction for the FA profile study. Besides, another 100 mL of each ruminal extraction were placed into a container for microbial composition analysis. The samples were feezed in appropiate falcon tubes (50 ml) in a industrial freezer.
Line (162): Fat content or ether extract content?
Corrected
Line (233-235): This activity did not influence the recognition of the characteristics observed by the panelists?
This is a standard protocol in sensory analyses
Results
Tables: Standardize the decimals of SED. Use two, three, or four decimal places.
We have revised all the tables. It is not possible to standarize decimals of SED. Since it is advisable to put one more decimal for the SED than for the corresponding mean and the mean of the different variables have different dimensions.
Lines (277-278; 289; 299; 308-309; 336; 344-345; 373-374; 403; 428-429; 440-441): This text is not relevant, delete it.
We have deleted the suggested sentences except for that of line 373-374 because in our oppinion it is necessary for contextualization of the presented results.
Discussion:
The discussion has lines that belong to the topic of results. Also, you wrote many texts comparing your results with other researchers. Finally, the explanation of the biological behavior (discussion) of its results is missing.
Lines (448-460): This text is irrelevant to discuss your data. This text can best be placed in the introduction.
These paragraphs have been deleted as requested.
Lines (462-465): How does this text correlate with the paragraph introduction about biohydrogenation (lines 461-462)?
Reviewer is correct. We have changed this sentence
Line (472): How is this statement true? Because CRPC concentrate has lower C18:0 fatty acid content.
We said that CPRC supplied greater amounts of C18 UFAs (unsaturated FA), mainly C18:1 cis-9 that, when are subjected to the biohydrogenation process, generated or are converted in C18:0.
However, it is known that the final reduction step of UFA to C18:0 is considered rate limiting and for this reason increasing concentration of other BH intermediates are also observed.
Conclusion:The conclusion must be improved.
The conclusion has been reworded
Round 2
Reviewer 3 Report
The authors worked on the specific comments; however, I felt a lack of improvement of the topics: material and methods, and discussion.
Although the evaluated parameters are important to the scientific community, the manuscript needs to be improved. The discussion needs to be better explored and in this you need to discuss the possible biological processes that interfered with your results when using CPRC.
Author Response
The authors worked on the specific comments; however, I felt a lack of improvement of the topics: material and methods, and discussion.
Although the evaluated parameters are important to the scientific community, the manuscript needs to be improved. The discussion needs to be better explored and in this you need to discuss the possible biological processes that interfered with your results when using CPRC.
Reviewer commented that it is a lack of improvement of material and methods section. We think we have answered all the specific comments regarding material and methods in the previous review. It is very difficult for us to improve it without further details.
We reworded some paragraphs to try to avoid repeating results in the discussion section (L444, L546, L575, L561)
Reviewer pointed out in the first review that the discussion section had many texts comparing the results with other researchers. We think that it is a common practice to justify the observed results comparing them with published results on the topic. Anyway, if there is any comparison with other researcher that the reviewer found redundant or not relevant please be more specific and we will proceed to delete them.
Rergarding the explanation of the results in a biological integrated fashion, in our oppinion in lines L465-482; L488-490; 490-494; 505-517; 520-527; 540-541; 551-554 we tried to explain the biological behaviour in terms of how differences in the experimental concentrate (basically fatty acid profile) can alter microbial populations and can modify BH process and ruminal metabolites. If there is any other biological behaviour that the reviewer thinks that is relevant and missing in the discussion section, we would appreciate very much if he/she could be more specific.
In the previous report the reviewer pointed out that CRPC partially replaced other ingredients in the concentrate. We agree that this cannot be obviated and we have included this assement in a new paragraph in L533-537 , as a biological process that could interfere with the results. Again, if there is any other relevant aspect that the reviewer believes that needs to be included we would appreciate very much if he/she could be more specific.